# Bryophytes Used in Folk Medicine: An Ethnobotanical Overview

**Riccardo Motti [1,\*], Anna Di Palma [2] and Bruna de Falco [3]**

1 Department of Agricultural Sciences, University of Naples Federico II, Via Università 100, Portici, 80055 Naples, Italy

2 Centre for Isotopic Research on Cultural and Environmental Heritage (CIRCE), Viale Carlo III, 153, 81020 San Nicola La Strada, Italy

3 Spanish Bank of Algae, University of Las Palmas de Gran Canaria, Muelle de Taliarte s/n, 35214 Telde, Spain

\* Correspondence: motti@unina.it

**Abstract:** Bryophytes are considered the oldest living plants of terrestrial habitats and the closest modern relatives of the ancestors of the earliest terrestrial plants. Bryophytes are found on all continents and occupy xeric to aquatic niches, with the greatest diversity and biomass in cool temperate regions. Despite the lesser popularity of these organisms, bryophytes have ethnopharmacological importance in different cultures of the world, especially in Chinese, Indian, and Native American medicine. Different bryophyte extracts and isolated compounds have shown anti-microbial, antiviral, and cytotoxic effects. The present overview aims to highlight the use of bryophytes for the treatment of common ailments in folk medicine around the world and to collect, analyze, and summarize the available literature on the pharmacological activity of the most used mosses and liverworts. Based on the literature review, 109 wild taxa of Bryophyta being used for ethnomedical purposes have been documented. Overall, 170 uses were recorded for the 109 taxa considered. Herbal remedies for skin and hair care are by far the most commonly reported (25.0%); antipyretic uses of bryophytes account for 12.2%, while taxa used as medicinal treatments for respiratory and gastro-intestinal systems amount to 12.1% and 9.9%, respectively.

**Keywords:** bryophytes; mosses; ethnomedicine; ethnobotany; ethnopharmacology; ethnobryology

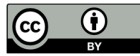

## 1. Introduction

Bryophytes are the only land plants with a life cycle featuring a haploid, branched gametophyte as a dominant generation, which alternates with unbranched diploid sporophytes. With the term bryophytes, approximately 24,000 species are counted, which are commonly referred to as liverworts, mosses, and hornworts, respectively belonging to three main divisions, i.e., Marchantiophyta (9000 species), Bryophyta (sensu stricto, 15,000 species), and Anthocerotophyta (300 species) [1]. Bryophytes are considered the oldest living plants of terrestrial habitats and the closest modern relatives of the ancestors of the earliest terrestrial plants [1]. Despite their limited competitive capabilities and low statures, these organisms are a highly successful group in comparison with other types of non-flowering land plants [2]. Bryophytes are found on all continents and occupy aquatic to xeric environments, with the greatest diversity and biomass in cool temperate regions where the single genus *Sphagnum* contains approximately 16% of the earth's carbon [3]. Ethnobotany was a term first suggested by John Harshberger in 1896 to delimit a specific field of botany and describe plant uses by aboriginal peoples [4]. Ethnobotany is a multidisciplinary science including taxonomy, nutrition, pharmacognosy, phytochemistry, ecology, and conservation biology [5]. Ethnobotany has also been constructed to include studies of those life forms traditionally considered to be plants, such as algae, lichens, and

mosses. Ethnopharmacology is the traditional knowledge of substances used as medicine by different cultural or ethnic groups and has recently become a major field of investigation in ethnobotany [6]. However, researchers have been interested in the observation, description, and experimental investigation of indigenous medicinal plant biology and use over the centuries [7]. Approximately 64 years ago, the term "ethnobryology" was introduced in a paper regarding the bryophytes utilized by the Gosuite people of Utah [8]. When compared with the ethnobotanical literature regarding vascular plants, the information about folk therapeutical applications of bryophytes is still scarce, especially in some geographical areas (e.g., Europe, Africa, South America). One of the causes of the lesser popularity of these organisms could be that they are smaller and less conspicuous than vascular plants [9] and the distinctive morphological characteristics are sometimes so reduced that identification is also very difficult, even with the aid of a microscope [10]. In the older literature, bryophytes were often confused with lichens, club mosses, or some vascular plants [11]. As highlighted by Alam [12], they also lie in less noticeable places and, hence, are often unnoticed by humans; therefore, bryophytes remain underexplored in many aspects including their medicinal value. Furthermore, bryophytes have no nutritional properties for humans; in fact, no data concerning their use as food have been found. In the last few decades, many researchers have studied the phytochemistry of these plants which have been shown to contain numerous potentially useful natural products (e.g., [13–16]). Different compounds isolated from bryophytes have demonstrated antimicrobial, antiviral, neuroprotective, and cytotoxic effects. Furthermore, they have shown positive effects on smooth muscles, such as the stomach, intestines, bladder, bronchioles, and uterus, in addition to playing a role in weight loss [17–19]. The role of ethnobotanical research is to avoid the loss of traditional knowledge concerning medicinal plant lore and simultaneously provide a basis for developing new drugs from phytochemical and biochemical research [20]. Although there are fewer ethnobotanical studies concerning bryophytes than vascular plants, it is important to emphasize that these plants have ethnopharmacological importance in different cultures of the world, especially in Chinese, Indian, and Native American medicine [8]. Harris [21] reported approximately 150 taxa used in ethnobotany around the world for different purposes and approximately 67 are cited for their medicinal properties. More recently, some reviews have taken into consideration the ethnomedicinal and pharmacological attributes of bryophytes [9,22,23], reporting between 50 and 60 taxa each. In this context, we reviewed the available literature in order to (a) highlight the use of bryophytes for the treatment of common ailments in folk medicine around the world and (b) collect, analyze, and summarize the available literature on the pharmacological activity of the most used mosses and liverworts.

## 2. Materials and Methods

Electronic literature searches were conducted using the online versions of Elsevier Journal Finder (https://elsevier.com, last accessed on 15 November 2022), Scopus (https://scopus.com, last accessed on 10 November 2022), and Google Scholar (https://scholar.google.com, last accessed on 27 November 2022), using the following keywords and connectors: "ethnobotany", OR "ethnomedicine", OR "phytotherapy", OR "ethnobryology", OR "mosses", OR "bryophytes", AND "medicinal", OR, "ailments", OR "ethnopharmacology". The criteria for article selection were defined a priori to avoid personal bias. Publications were filtered for English languages, duplicates, document type (no patents), and full-text availability. A simple evaluation of both titles and abstracts was carried out for every result in relation to the use of bryophytes in folk medicine. The articles filtered had their abstracts fully read in order to further reveal the real interest of the review article and filter out non-applicable studies. The results thus obtained had their full contents read and evaluated. Only articles containing specific references to the use of bryophytes were included. Besides the articles gathered from the online databases, further papers were selected from the references cited by the studies previously collected. To classify the ailments, we used a symptom-based nosological approach, a categorization that

is widely used in ethnobotanical research (e.g., [24,25]). The antifungal, antimicrobial, and antidotal properties of some species have been added to these categories. The nomenclature follows the World Flora Online [26]. Abbreviations of authors are standardized according to Brummitt and Powell [27], as recommended by Rivera et al. [28]. The obtained data are presented in charts and tables.

## 3. Results and discussion

Based on the literature review, 109 wild taxa of Bryophyta being used for ethnomedical purposes have been documented (Table 1). The taxa belong to 44 families and 69 genera. Polytrichaceae is the most frequently cited family (11.2%), followed by Pottiaceae (9.2%), Mniaceae (7.1%), and Marchantiaceae and Sphagnaceae (both 6.1%). The most species-rich genus is *Sphagnum* (six species), followed by *Marchantia* and *Pogonatum* (both four species). Mosses accounted for 77.5% of total citations, while liverworts accounted for the remaining 22.5%.

**Table 1.** Identified mosses and liverworts used as traditional herbal remedies. (Au = Australia; Bo = Bolivia; Br = Brazil; Ca = Canada; Cn = China; De = Germany; Ec = Ecuador; Eg = Egypt; Es = Spain; Gt = Guatemala; In = India; It = Italy; Mx = Mexico; NZ = New Zealand; Pe = Peru; Ph = the Philippines; Us = USA; Gb = United Kingdom). (N.A. = Not Available).

| Species (Family) | Preparation (When Reported) | Medical Uses | Country | Ref. | Phytochemical Studies |
|---|---|---|---|---|---|
| *Aerobryidium filamentosum* (Hook.) M. Fleisch. (Meteoriaceae) | | Clears heat and relieves toxicity. Burns. | Cn | [21] | N.A. |
| *Aerobryum lanosum* (Mitt.) Mitt. (Brachytheciaceae) | The whole plant boiled in goat urine is applied externally | Burns. | In | [29–31] | N.A. |
| *Amblystegium serpens* (Hedw.) Schimp. (Amblystegiaceae) | | External injuries and bleeding. | Cn | [21] | [32] |
| *Archilejeunea ludoviciana* (De Not. ex Lehm.) Gradst. & P.Geissler subsp. *porelloides* (Spruce) Gradst. & P.Geissler (Lejeuneaceae) | Decoction | Chest pains. | Bo | [33] | N.A. |
| *Atrichum undulatum* (Hedw.) P. Beauv. (Polytrichaceae) | | Anti-cancer. | Cn | [34] | [17,35] |
| *Barbula* sp. (Pottiaceae) | Infusion | Colds, fever, and body aches. | Ph | [36] | N.A. |
| *Barbula indica* (Hook.) Spreng. (Pottiaceae) | | Menstrual pains and intermittent fever. | In | [31] | [14,37] |
| *Barbula unguiculata* Hedw. (Pottiaceae) | | Fire sickness, fever, and body aches. | IN, U | [21,31,38] | [32] |
| *Bartramia ithyphylla* Brid. (Bartramiaceae) | | Suppress fear, calms nerves. Irregular heartbeat, epilepsy, apoplexy. | Cn | [34] | [39] |
| *Brachythecium* sp. (Brachytheciaceae) | | To treat fever and for detoxification. | Cn | [21] | N.A. |
| *Braunia secunda* (Hook.) Bruch & Schimp. (Hedwigiaceae) | Boiled in water, then used as a wash for the head. | To relieve headaches. | Mx | [40] | N.A. |

| | | | | | |
|---|---|---|---|---|---|
| *Bryum* sp. (Bryaceae) | The moss was rubbed into a paste and applied as a poultice. | Healing wounds, burns, and bruises or as padding under splints in setting fractures. Antifungal. | Cn, Ph, Us | [21,29,41–43] | N.A. |
| *Bryum argenteum* Hedw. (Bryaceae) | | Detoxifying antidote. Nose inflammation, antipyretic. Antibacterial, antifungal. Dysentery | Cn | [21,29,30,44–47] | [48] |
| *Bryum capillare* Hedw. (Bryaceae) | | Fire sickness, fever, and body aches. | Us | [21] | N.A. |
| *Climacium dendroides* (Hedw.) F.Weber & D.Mohr (Climaciaceae) | | Clears heat, removes moisture, relaxes muscle. Rheumatism and bone and muscle pain. | Cn | [21] | N.A. |
| *Conocephalum* sp. (Conocephalaceae) | | Skin diseases. Antipyretic. Antimicrobial and antifungal. | In | [49] | |
| *Conocephalum conicum* (L.) Underw. (Conocephalaceae) | | Cuts, swollen tissue, scalds, burns, and fractures. Snake bites, gallstones. Jaundice, as antimicrobial, antifungal, and antipyretic. | Cn, IN, It | [21,30,46,50,51] | [52] |
| *Cratoneuron filicinum* (Hedw.) Spruce (Amblystegiaceae) | | Calming and soothing; heart problems. | Cn | [21,29] | [15,18] |
| *Dawsonia superba* Grev. (Polytrichaceae) | | Diuretics. Hair growth stimulation. Cold. | Ca, Cn, Ph | [29,38,46,53,54] | N.A. |
| *Dendropogonella rufescens* (Schimp.) Britt. (Cryphaeaceae) | | Discomfort of women after childbirth. Body and bone pain. Kidney and lung health. Diabetes-related ailments. Blindness. Appetizer. | Mx | [55] | N.A. |
| *Dicranium bonjeannii* De Not. (Dicraniaceae) | | Absorbent. | Ca, Us | [56,57] | |
| *Dicranum majus* Turner (Dicranaceae) | | Clears lungs and stops cough. | Cn | [21] | [58] |
| *Diplophyllum* sp. (Scapaniaceae) | | Anti-leukemic. | Us | [59] | [60] |
| *Ditrichum pallidum* (Hedw.) Hampe (Ditrichaceae) | | Convulsions, particularly in infants | Cn, In | [21,31] | N.A. |
| *Dumortiera* sp. (Marchantiaceae) | | Antibacterial. | In | [49] | |
| *Dumortiera hirsuta* (Sw.) Nees (Marchantiaceae) | | Antibacterial. | Ph | [38] | [52,61,62] |

| | | | | | |
|---|---|---|---|---|---|
| *Entodon compressus* (Hedw.) Müll. Hal. (Entodontaceae) | | Diuretic and to reduce swelling. | Cn | [21,46] | N.A. |
| *Entodon flavescens* (Hook.) A. Jaeger (Entodontaceae) | Leaf juice as drops. | During cold for earache. | Cn | [29,38] | N.A. |
| *Fissidens* sp. (Fissidentaceae) | | Antibacterial. Sore throat. | Cn | [21] | N.A. |
| *Fissidens nobilis* Griff. (=F. japonicus Dozy and Molk.) (Fissidentaceae) | | Diuretics. Hair growth stimulation. Tonics. Antibacterial. Burns. Choloplania. | Cn, Ph | [21,29,38] | N.A. |
| *Fissidens pellucidus* Hornsch. (=Fissidens flexinervis Mitt.) (Fissidentaceae) | Decoction. | Digestive. | Bo | [33] | N.A. |
| *Fontinalis antipyretica* Hedw. (Fontinalaceae) | | Antipyretic and for detoxification. | Cn | [21] | N.A. |
| *Frullania* sp. | | Antimicrobial. | In | [49] | |
| *Frullania ericoides* (Nees ex Mart.) Mont. (Jubulaceae) | The whole plant is made into a paste and roasted in coconut oil. | To get rid of head lice (*Pediculus humanus*) and nourishment of hair. | In | [63] | N.A. |
| *Frullania tamarisci* (L.) Dumort. (Jubulaceae) | | Antiseptic. | Cn | [21,30] | N.A. |
| *Funaria hygrometrica* Hedw. (Funariaceae) | | Pulmonary tuberculosis. Hemostatic, bruises, skin infections, athlete's foot. Blood vomiting. Light sedative. Nose inflammation and sinusitis. Alopecia. Bronchitis, tonsillitis, pneumonia, and fever. | Cn, De | [21,29,30,45,46] | N.A. |
| *Haplocladium microphyllum* (Hedw.) Broth. (Thuidiaceae) | | Erysipelas, sores, bladder, mammary glands, and middle ear inflammation. Postpartum infections. Cystitis. | Cn | [21,29,46] | N.A. |
| *Herbertus* sp. (Herbertaceae) | | Antiseptic. Diarrhea. Cough. | Ph | [38,51] | N.A. |
| *Hylocomium splendens* (Hedw.) Schimp (Hylocomiaceae) | As a poultice. | Sores. | Ca, It | [64,65] | N.A. |
| *Hyophila attenuata* Broth. (Pottiaceae) | Decoction administered with a pinch of pepper powder. | Cold, cough. Neck pain. | Cn | [29,30] | N.A. |
| *Hyophila involuta* Jaeger (Pottiaceae) | | Cuts and wounds. | In | [21] | N.A. |
| *Isopterygium tenerum* (Sw.) Mitt. Hypnaceae) | Decoction. | Rheumatism. | Bo | [33] | N.A. |

| | | | | | |
|---|---|---|---|---|---|
| *Lembophyllum clandestinum* (Hook.f. & Wilson) Lindb. (Lembophyllaceae) | As a wrapper or absorbent. Steeped in water. | For care and nursing of babies and in sanitary napkins. | NZ | [21] | N.A. |
| *Leptodictyum riparium* (Hedw.) Warnst. (Amblystegiaceae) | | Venereal diseases. Antipyretic. Choloplania. Urinary tract disorders. | Cn | [21,29,30,46] | N.A. |
| *Leucobryum bowringii* Mitt. (Dicranaceae) | Paste of leaf tips mixed in a cup of Phoenix sylvestris. | Body pains. | Cn | [30] | N.A. |
| *Leucodon secundus* (Harv.) Mitt. (Leucodontaceae) | | Hemostatic. Bruises, swelling, and pains. Headache. Stomach ache. | Cn, I | [21,66] | N.A. |
| *Lunularia cruciata* (L.) Lindb. (Lunulariaceae) | | Kidney ailments. Faintings. | Pe | [49,67] | [68] |
| *Marchantia* sp. (Marchantiaceae) | | Antipyretic. Mouth sores. Liver diseases. Pulmonary tuberculosis. Skin diseases. | In, Us | [49,69] | [70] |
| *Marchantia convoluta* Gao et K.C. Zhang (Marchantiaceae) | | Hepatitis. Antipyretic. Gastric intolerances. | In | [71] | [68,72] |
| *Marchantia paleacea* Bertol. (Marchantiaceae) | | Skin tumefaction. | | [73] | [19,74] |
| *Marchantia palmata* Nees (Marchantiaceae) | | Hepatitis. Antipyretic. Burn, boils, and abscesses. | Cn, In | [29,75] | [10,19,70] |
| *Marchantia polymorpha* L. (Marchantiaceae) | | Diuretics Liver ailments. Pulmonary tuberculosis. Cardiovascular diseases, stones in the bladder. Skin inflammations, insect bites, boils, abscesses and eruption of pimples, fractures, poisonous snake bites, burns, scalds, and open wounds. | Br, Cn, In, Es | [21,57,75–81] | [10,19,61,82] |
| *Meteoriella soluta* (Mitt.) S.Okamura (Pterobryaceae) | | Light calming. External, gastrointestinal, and lung bleeding. | Cn | [46] | N.A. |
| *Mnium* sp. (Mniaceae) | Poultice | Burns, bruises, and wounds. | Ph | [38] | N.A. |
| *Octoblepharum albidum* Hedw. (Dicranaceae) | Decoction | Headache, fever, and body aches. Sedative. | Bo, Us | [21,33,38] | [83,84] |
| *Oreas martiana* (Hoppe and Hornsch.) Brid. | | Wounds. Epilepsy. Menstrual disorders. | Cn | [21,29,30,46] | N.A. |

| Species (Family) | Preparation | Uses | | | |
|---|---|---|---|---|---|
| (Dicranaceae) | | Neurasthenia. Rheumatism. Stomach pain. Sedative. | | | |
| *Orthostichopsis tortiilis* (Müll. Hal.) Broth. (Pterobryaceae) | | Cuts. Stomach ache. Snake bites. | Ec | [21] | N.A. |
| *Pallavicinia* sp. (Pallaviciniaceae) | | Antimicrobial. | Ph | [38] | [85] |
| *Palustriella commutata* (Hedw.) Ochyra (Amblystegiaceae) | | Antipyretic. Detoxification. | Cn | [21] | N.A. |
| *Pellia* sp. (Pelliaceae) | The juice was drunk or the plant chewed. | Sore throat. | Ca | [86] | |
| *Philonotis* sp. (Bartramiaceae) | | Burns. Adenopharyngitis. Antipyretic. Antidote. | Ph, Us | [21,29,30] | N.A. |
| *Philonotis fontana* (Hedwig) Bridel (Bartramiaceae) | | Antipyretic. Drawing out toxins. Sore throat. Diuretic, urinary obstructions. | Cn | [21,30] | N.A. |
| *Plagiochasma appendiculatum* Lehm. & Lindenb. (Rebouliaceae) | Paste. | Burns, boils, blisters, wound healing. | In | [51,77,87–89] | [61,90] |
| *Plagiochasma rupestre* (Forster) Steph. (Rebouliaceae) | | Kidney ailments. Faintings. | Pe | [67] | [90] |
| *Plagiochila* sp. (Plagiochilaceae) | | Anti-leukemic. Antimicrobial. | Cn, In | [30,38,51] | [91] |
| *Plagiochila beddomei* Steph. (Plagiochilaceae) | | Wound healing | In | [31] | [10,92] |
| *Plagiomnium acutum* T.Koponen (Mniaceae) | | Hemostatic for nose, gastrointestinal tract, teeth, gumsloods. Spitting and coughing blood. Blood in urine or stool; uterine bleeding. | Cn | [21,29,45,46] | [22,93] |
| *Plagiomnium cuspidatum* (Hedw.) T. Kop. (=Mnium cuspidatum Hedw.) (Mniaceae) | | Hemostatic. | Ph | [38] | [93,94] |
| *Plagiomnium insigne* (Mitt.) T.J. Kop. (Mniaceae) | Poultice. | Boils. Breast abscesses in women. Swellings. | Ca | [21] | [95] |
| *Plagiomnium* sp. (Mniaceae) | | Infections and swellings. | Ph | [38,88] | N.A. |
| *Plagiopus oederi* (Brid.) Limpr. (Bartramiaceae) | | Sedative, epilepsy, apoplexy. Cardiovascular diseases. | Cn | [21,29,30,46] | N.A. |
| *Pleurochaete squarrosa* (Brid.) Lindb. (Pottiaceae) | As tea. Boiled and then placed on a wound. | Stomach ache. Wound healing. | Mx | [40] | N.A. |
| *Pogonatum* sp. | | Diuretic. Hair growth. | Cn | [21,29] | N.A. |

| | | | | | |
|---|---|---|---|---|---|
| (Polytrichaceae) | | | | | |
| *Pogonatum cirratum* Bridel (Polytrichaceae) | | Cardiovascular diseases. Neurasthenia. Sedative. Hemostatic. | Cn | [21] | [96] |
| *Pogonatum inflexum* (Lindb.) Sande Lac. (Polytrichaceae) | | Heart palpitations, insomnia. Wound healing. | Cn | [21] | N.A. |
| *Pogonatum macrophyllum* Dozy & Molk. (Polytrichaceae) | | Anti-inflammatory. Antipyretic. Diuretic, laxative. Hemostatic. | Ph | [38] | N.A. |
| *Pogonatum microstomum* (R.Br. ex Schwaegr.) Brid. (Polytrichaceae) | | Gallstones. | Cn | [21] | [97] |
| *Polytrichastrum alpinum* (Hedw.) GL Sm. (Polytrichaceae) | | Calming. Cough. Hemostatic. | Cn, Es | [21,98,99] | [100] |
| *Polytrichum* sp. (Polytrichaceae) | Thallus powder with oil. Fleshy paste. | Healing burns, bruises, wounds, and other skin ailments. Diuretic. Antipyretic, anti-inflammatory, and antidotal. Hemostatic. Gallbladder and kidney stones. Hair care. | Cn, In, Ph Us | [21,38,41,76,77] | N.A. |
| *Polytrichum commune* Hedw. (Polytrichaceae) | | Healing burns, bruises, wounds, and other skin ailments. Antipyretic. Common cold. Diuretic. Anti-inflammatory and antidotal. Hemostatic. Gallbladder and kidney stones. To speed up the birth of a baby during childbirth. To strengthen hair. | Ca, Cn, DeEc, Gt, In, GB | [21,43,54,56,57,101,102] | [103] |
| *Polytrichum juniperinum* Hedw. (Polytrichaceae) | | Prostate diseases. Urinary difficulties. Sores, boils, and swelling. | Ca, In, GB | [21,38,51] | N.A. |
| *Reboulia* sp. (Aytoniaceae) | | Skin problems. Haemostatic. | In | [49] | |
| *Reboulia hemisphaerica* (L.) Radd (Aytoniaceae) | | Hemostatic. Wounds and bruises. | Cn | [30] | [90] |
| *Rhizomnium glabrescens* (Kindb.) T.J. Kop. (Mniaceae) | | Boils, blood blisters. Breast abscesses in women. | Ca | [21] | N.A. |

| | | | | | |
|---|---|---|---|---|---|
| *Rhizomnium punctatum* (Hedw.) T.J. Kop. (Mniaceae) | | Swellings. | Us | [21] | [93] |
| *Rhodobryum giganteum* (Schwaegr.) Par. (Bryaceae) | | Antipyretic, Diuretic. Sedative. Cardiovascular disease. High blood pressure. Red eye. Cuts. | Cn | [21,29,30,46,104] | [105,106] |
| *Rhodobryum roseum* (Hedw.) Limpr. (Bryaceae) | | Cardiovascular diseases. High cholesterol. Sedative. | Cn, In | [31,107] | N.A. |
| *Riccardia* sp. (Aneuraceae) | | Anti-leukemic. | In, Ph | [38,51] | N.A. |
| *Riccia* sp. (Ricciaceae) | The thallus is ground and mixed with jaggery. | Ringworm in children. | In | [49,76,77] | N.A. |
| *Sematophyllum adnatum* (Michx.) E. Britton (Sematophyllaceae) | | To prepare medicinal teas. | Mx | [40] | N.A. |
| *Sphagnum* sp. (Sphagnaceae) | | Burns, wounds. Eye diseases. Surgical dressing. Sores. | Au, Ca, Cn, DeEc, In, Nz, Us, Gb | [21,42,43,57,76,108–110] | N.A. |
| *Sphagnum girgensohnii* Russow (Sphagnaceae) | | Surgical dressing. | Cn | [21,29,43,46] | N.A. |
| *Sphagnum magellanicum* Brid. (Sphagnaceae) | | Surgical dressing. Diapers. | Ca, Cn | [29,43,111] | [112] |
| *Sphagnum palustre* L. (Sphagnaceae) | | Surgical dressing. Eye diseases. | Cn | [21,29,46,113] | [112] |
| *Sphagnum sericeum* C. Mull. (Sphagnaceae) | | Dressing wounds, with anti-microbial properties, insects bites, scabies, acne. Hemorrhoids. Eye diseases. | Cn, Ph | [38,46,102] | N.A. |
| *Sphagnum squarrosum* Crome (Sphagnaceae) | | Surgical dressing. | Cn | [21,29,43,46] | N.A. |
| *Sphagnum teres* (Schimp.) Angstrom (Sphagnaceae) | | Eye diseases. Sedative | Cn | [29,46] | N.A. |
| *Targionia hypophylla* L. (Targionaceae) | Ground into a paste and mixed with coconut oil. | Scabies, itches, and other skin diseases. | In | [63] | [90] |
| *Taxiphyllum taxirameum* (Mitt.) M. Fleisch. (Hypnaceae) | | Anti-inflammatory. Hemostatic. Wound healing. | Cn, I | [21,29,31,46] | N.A. |
| *Tetraplodon mnioides* Bruch & W.P.Schimper (Splachnaceae) | | Sedative. Stroke. Epilepsy. | Cn | [21,46] | N.A. |
| *Thuidium cymbifolium* Dozy & Molkenboer (Thuidiaceae) | | Burns. | Cn | [21] | N.A. |

| | | | | | |
|---|---|---|---|---|---|
| *Thuidium schistocalyx* (Müll. Hal.) Mitt. (Thuidiaceae) | Decoction. | Headaches. | Bo | [33] | N.A. |
| *Timmiella* sp. (Pottiaceae) | | Cuts and swellings. | Eg | [21] | N.A. |
| *Trichosteleum papillosum* (Hornsch.) A.Jaeger (Sematophyllaceae) | Decoction. | Rheumatism. | Bo | [33] | N.A. |
| *Weisia controversa* Hedwig (Pottiaceae) | | Clears heat and relieves toxicity. Nose inflammation and sinuses. | Cn | [21] | N.A. |
| *Weisia viridula* (L.) Hedw. (Pottiaceae) | | Cold. Antipyretic. | Cn | [21,29,38] | N.A. |
| *Wiesnerella denudata* (Mitt.) Stephani (Wiesnerellaceae) | | Anti-leukemic. | In | [51] | [114] |

From the analyses carried out on a global scale (Figure 1), we found that China had the highest number of records of bryophytes used for medicinal purposes (54), followed by India (23), the Philippines (15), and the USA (11). At the continental scale, Asia had more than half of the total records (98), followed by America (48) and Europe (9).

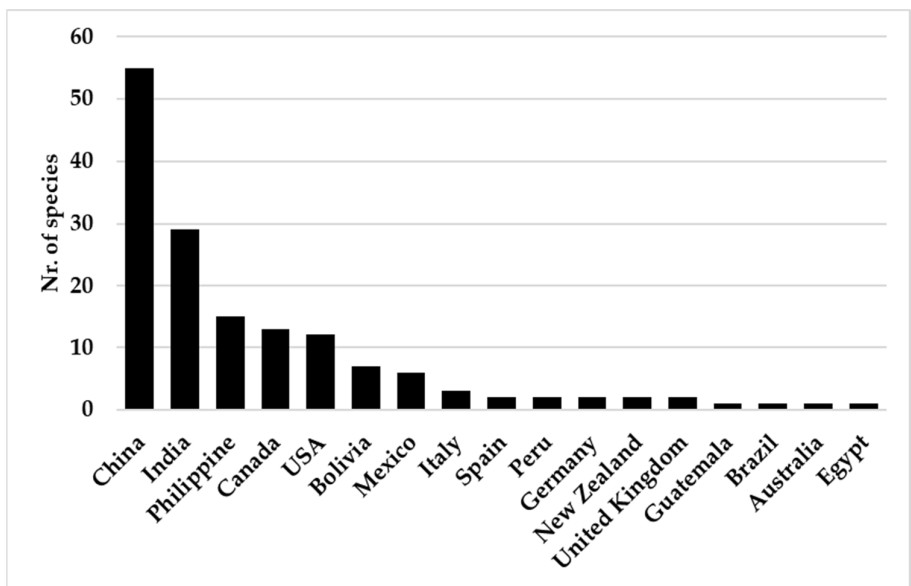

**Figure 1.** Number of species used for medicinal purposes for each country.

In all, 21 species were reported to have three or more uses for different ailment categories. *Funaria hygrometrica* had five different use reports, followed by *Bryum argenteum*, *Conocephalum conicum*, *Dendropogonella rufescens*, *Fissidens nobilis*, and *Orthostichopsis tortipilis*, which was used for the treatment of four different ailment categories. On the basis of our literature findings regarding phytochemical properties, validating studies were obtained in approximately 37% of cases.

Overall, 170 uses were recorded for the 102 taxa considered. Herbal remedies for skin and hair care were by far the most commonly reported (25.0%) (Figure 2); antipyretic uses of bryophytes accounted for 12.2%, while taxa used as medicinal treatments for respiratory and gastro-intestinal systems amounted to 12.1% and 9.9%, respectively.

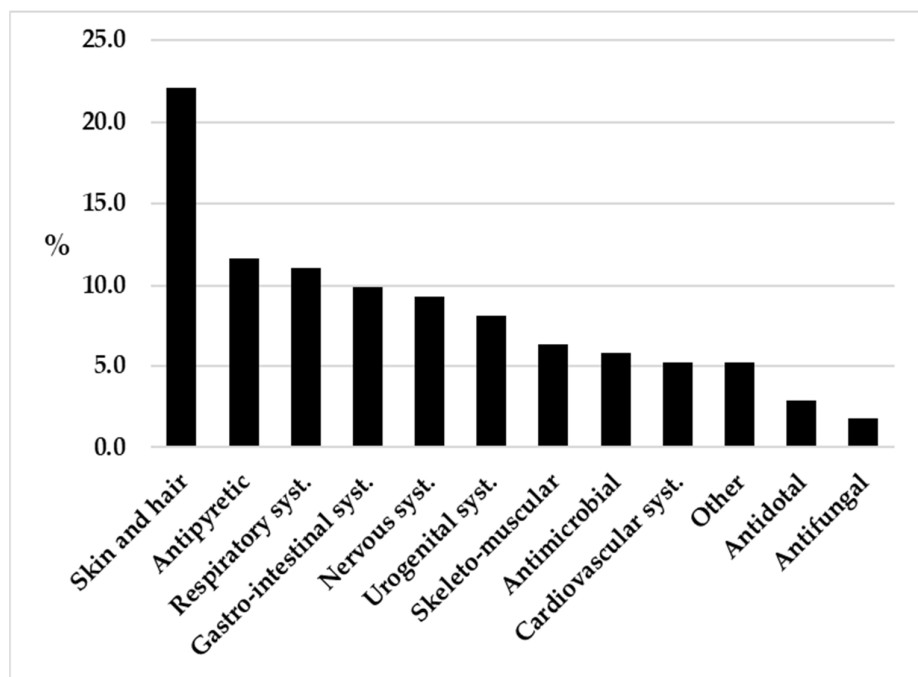

**Figure 2.** Ailments categories for which bryophytes are used.

Different preparations and application processes of medicinal bryophytes were mentioned for topical uses. As shown in Figure 3, the majority of these remedies involved treatments for healing wounds, burns, boils, abscesses, bruises, and swellings.

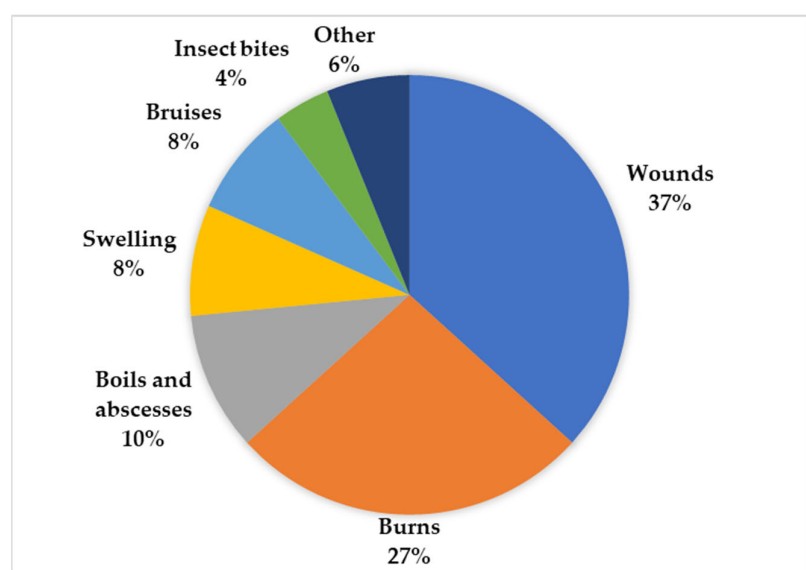

**Figure 3.** Topical uses of bryophytes for skin problems.

Among the mosses, the *Sphagnum* genus was the one mainly used for skin care and wound healing. The Flora Lapponica [115] reported that *Sphagnum palustre* was used by Sámi women as diapers to absorb urine and as menstrual pads. Linnaeus also reported that in Lapland, babies slept in leather cradles without swaddling clothes, but were protected against the most intense cold by dried bog moss lined with the hair of reindeer [116]. Occasional use of peat moss as a dressing material occurred independently in various regions of the world [117]. The use of *Sphagnum* as diapers by the Natives Peoples of

North America [118] is also well known. Alaskan Natives made ointment from *Sphagnum* leaves mixed with tallow and grease to treat cuts [20]. During World War I, dried *Sphagnum* was used in Britain, Canada, and Germany as a replacement for bandages [119,120]. Furthermore, during the Russo–Japanese War, *Sphagnum* mosses were used by the Japanese as a first-aid dressing on a large scale [121]. Nichols [122] reported that *Sphagnum* pads, used for bandages in World War I, could absorb up to 22 times their mass, making them 5–6 times as absorptive as cotton pads. The use of *Sphagnum* for wounds and other skin disorders is due to its fast absorption capability and various phenolic compounds that are known to have antimicrobial properties [16]. According to Painter [123], wound healing may be promoted not only because of the absorptivity property but also due to a Maillard reaction between the free amino groups of collagen and reactive carbonyl groups in a soluble glycuronoglycan ('sphagnan') containing residues of d-lyxo-5-hexosulopyranuronic acid. *Sphagnum* wound dressings combat infection by immobilizing bacterial cells and depriving them of their nutrients; all pathogens, irrespective of identity, are subject to the action of sphagnan [124]. *Sphagnum* species were also used in postoperative dressings [22]. *Sphagnum sericeum* Hal. has also been used for insect bites, acne, scabies, hemorrhoids, and to treat eye diseases [9,38,46,102]. Other bryophytes are involved in the treatment of dermatological disorders: Native Americans, for instance, used a mixture of moss ash and honey as a disinfectant for wounds [125]. The liverwort *Plagiochasma appendiculatum* is widely used in India for burns, boils, and blisters and its extract shows wound healing and potent antioxidant activity [87–89]. The tribals of Melghat area and the Gaddi Tribes of Kangra Valley in their herbal health care systems used the fine paste of thoroughly washed mature thalli with the female receptacle of *P. appendiculatum* applied externally for skin disease treatment and for the treatment of skin eruptions caused by the hot sun during summer [77,89]. The efficacy of this liverwort for wound healing may be due to its action against dermatophyte which, in turn, can be correlated to the effect on antioxidant enzymes [87]. According to Singh et al. [88] and Glime [126], *P. appendiculatum* was found to have an inhibitory effect against *Pseudomonas aeruginosa, Klebsiella pneumoniae,* and *Candida albicans.* The acetone-soluble *P. appendiculatum* extracts showed 21.7% [127] growth inhibition in *C. albicans* at 10% green talli extract and 86.8% at 100% concentration. The chloroform fraction showed an inhibition zone (in relation to the standard drug erythromycin) in percentages of 92.5% against *Pseudomonas aeruginosa* and 87.4% against *Klebsiella pneumoniae* [88].

Among liverworts, some species of the *Marchantia* genus (*M. paleacea, M. palmata, M. polymorpha*) have been used for skin care. *Marchantia* species are rich in flavonoid, tannins, terpenes, oils, phenolic compounds, and bis-bibenzyls. The flavonoids include quercetin, luteolin, and apigenin and their glycosides [128]. In more detail, the main secondary metabolites of the methanol extract of *M. polymorpha* were cyclic and acyclic bis-bibenzyls, Marchantin A, Marchantin B, Marchantin D, Marchantin E, Perrottetin F, and Paleatin B [129]. The main constituents of the ether extract identified by GC-MS were isoprenoid compounds, including thujopsene, acoradiene, β-chamigrene, cuparene, β-himachalene, γ-cuprenene, and α-chamigren-9-one [130]. Marchantin A and the macrocyclic bis (bibenzyl) plagiochin E are the main constituents of *M. polymorpha* and showed antifungal, antimicrobial, and anti-cancer activities [70,131,132]. Plagiochin E, isolated from *M. polymorpha*, has efficacy at a dose of 100 μg/mL for the formation of chitin cell walls of *Candida albicans* [133]. Marchantin A, B, and D, paleatin B, and perrottetin F compounds also showed cytotoxicity against leukemic KB and P-388 cells at 3.7-20 μM dose concentrations [9,30].

Moreover, highly evolved liverworts belonging to the Marchantiaceae family produce phytosterols, such as campesterol, stigmasterol, and sitosterol; almost all liverworts elaborate a-tocopherol and squalene [10].

*Plagiochila beddomei* is widely used in folk phytotherapy in India to promote wound healing. Manoj and Murugan [92] used animal models of skin wounds to demonstrate that methanolic and aqueous extracts from *P. beddomei* promoted the formation of

granulation tissue, collagen production, and angiogenesis. Moreover, this liverwort showed potent antimicrobial activity against a wide group of bacteria such as *Staphylococcus aureus*, *Klebsiella pneumoniae*, and *Escherichia coli* and fungi such as *Candida albicans* [30,92,133]. In addition to its antimicrobial properties, *P. beddomei* could be considered a valuable source of bioactive constituents, which are expected to have a protective action against peroxidative damages in living systems in relation to aging and carcinogenesis [134].

*Bryum* species and *B. argenteum* in particular are used in folk medicine to heal wounds and for the treatment of burns and bruises (e.g., [21,29,46,47]). *B. argenteum* with a confirmed flavonoid content was reported to be active against different bacterial and fungal strains [135-137]. In particular, *B. argenteum* showed in vitro antimicrobial effects of different extracts against *Bacillus cereus*, *Escherichia coli*, *Klebsiella pneumoniae*, *Pseudomonas aeruginosa*, *Staphylococcus aureus*, *Enterococcus faecalis*, *Enterobacter aerogenes*, and *Proteus mirabilis.* The highest activity was shown against *E. coli* and *S. aureus* (MICs of 30–70 g/mL) [32]. This species has also been used as an antidotal, antipyretic, and antirhinitic treatment [51].

*Conocephalum conicum* is widely used to treat cuts, swollen tissue, burns, and fractures (e.g., [21,30,46,50,138]. This taxon has at least three chemo-types in Japanese species. One of them (type-I) showed sabinene as a major compound; the other two types characteristically contained bornyl acetate and methyl cinnamate as major constituents, respectively [139]. The constituents of European *C. conicum* and Japanese species of type-I were very similar except for the presence of a large proportion of a sesquiterpene alcohol, conocephalenol, in the former specimens [114]. According to the literature, mainly monoterpenic esters, sesquiterpene lactones, and phenethyl glycosides have been isolated from this liverwort, but not macrocyclic bis-bibenzyls [131]. The antimicrobial activity of *C. conicum* is controversial in the scientific literature. While some authors report that some extracts of these species have antibacterial activities against tested bacterial strains, such as *Pseudomonas aeruginosa* and *Klebsiella pneumoniae* (e.g., [88,140]), other studies show that this species has no antimicrobial properties [131,141]. Recently, three new sesquiterpenoids, namely, (1Z,4E)-lepidoza-1(10), 4-dien-14-ol (1), rel-(1(10) Z,4S,5E,7R)-germacra-1(10), 6 diene-11,14-diol (2), and rel-(1(10) Z,4S,5E,7R)-humula-1(10),5-diene-7,14-diol (3), have been isolated from the liverwort *C. conicum* [142].

Although in ethnobotanical studies, *Lunularia cruciata* is cited to treat different ailments (kidney diseases or as a remedy for faintings), this species possesses one of the most significant antibiotic activities among the bryophytes tested (e.g., [143,144]). Chemical analysis of *L. cruciata* has revealed the presence of lunularin, lunularic acid, and bisbibenzyls and their derivatives (e.g., perrottetin F, riccardins), luteolin-7-O-glucoside and quercetin, which have antimicrobial, antioxidant, cytotoxic, and cardiotonic activities [145,146]. *Amblystegium serpens*, reported to cure external injuries and bleeding, shows both antimicrobial and antiproliferative activities [140].

According to Russell [141], the occurrence of antibiotic substances appears to be more frequent in hepatics' 88% of liverworts demonstrated antibiotic activity while only 33% of mosses demonstrated antimicrobial activity.

*Octoblepharum albidum* is commonly used to treat headaches, fever, and body aches and as a sedative [21,33]. In the current literature, it is reported that *Octoblepharum albidum* inhibits $\alpha$-glucosidase, $\alpha$-amylase, and the aldose-reductase. Moreover, this species also showed hypoglycaemic and anti-hyperlipidemic properties [83]. *O. albidum* could also be used as a source of aminoglycosides derivatives that have antibacterial activities [84]. *Cratoneuron filicinum*, mainly used as a calming and soothing remedy and for heart problems, has highly radical scavenging effects [15]. Furthermore, this species also shows significant antibiotic activity against *Escherichia coli*, *Bacillus cereus*, and *Micrococcus flavus* [18]. *Rhodobryum giganteum* is used for the treatment of a wide range of complaints and *R. roseum* has long been used to treat cardiovascular disorders and nerve problems in China [22]. Ergosta-7,22-dien-3β,5α,6β-triol, ursolic acid, succinic acid, uracil, palmitic acid, quercetin,

nonacosane, β-sitosterol, and daucosterol are the main components of *R. giganteum* [105]. Rhopeptin A was isolated as the first cyclopentapeptide from this moss. This compound consists of proline, phenylalanine, and 3-hydroxyproline ring-bonded amino acid residues connected to a tyrosine fragment via an ether bridge [147]. *Sphagnum palustre* has been used as a surgical dressing but recent studies [148] showed that an extract of this species decreased estrogen biosynthesis by inhibiting aromatase activity. Aromatase inhibition can be considered one of the most efficient treatments in breast cancer therapy, especially for postmenopausal women [149,150].

As highlighted by several authors (e.g., [8,57,70,89,102]), an ancient method of determining the medicinal properties of plants was based on the concept of Paracelsus "doctrine of signatures", which could be stated as form summarizes function—the physical characteristics of plants reveal their therapeutic value. This belief has played a major role in the use of bryophytes, especially liverworts, in herbal medicine. *Marchantia polymorpha* and *Conocephalum conicum*, for example, are used to cure hepatic disorders as their structures resemble the lobes of livers [89]. *Marchantia palmata and M. polymorpha* are used to treat boils and abscesses as the developing archegonium of these liverworts emerges as a protuberance that resembles a tiny boil [42,151]. *Polytrichum commune* bears hairy calyptra and an oil extract from the calyptra has been used to strengthen and beautify women's hair [8,102]. Due to the long-stemmed and hair-like thallus of *Frullania ericoides*, this liverwort is applied to hair-related afflictions by tribal peoples of South India [63]. *Riccia* spp. is applied externally to the skin for the treatment of ringworm. The *Riccia* rosettes resemble, in fact, the characteristic rings that appear on the skin and, hence, are believed to cure these skin eruptions [151]. *Targionia hypophylla* is used in Kerala to cure skin ailments due to the resemblance of the thallus of this liverwort to the rough surface of the diseased part [8].

## 4. Conclusions

The studies included in this review demonstrate that bryophytes are used extensively in popular herbal medicines worldwide. Their use is closely linked to both the local flora and traditional knowledge. As with higher plants, the correct identification and in-depth knowledge of the species are of fundamental importance for their use. Furthermore, the problems of potentially toxic elements (e.g., pollution, pesticides) or dangerous microorganisms should be taken into consideration when using mosses and liverworts as natural medicines. In the last few decades, the ethnomedicinal importance of bryophytes has received increased attention and many chemicals and secondary metabolites have been isolated from different liverworts and mosses and different therapeutic activities have been studied. Some mosses and liverworts have been revealed as sources of new antibacterial and antifungal agents. Recently, new cultivation techniques have enabled the cultivation of bryophytes at large scales; thus, it is crucial to expand our knowledge about chemical and biological properties, including the toxicity evaluation of the bryophytes extracts and phytochemicals. In addition, pre-clinical and clinical studies are needed to better clarify the mechanism of action of bioactive compounds and confirm the potential of these alternative therapies.

**Author Contributions:** Conceptualization, R.M.; methodology, R.M.; investigation, R.M.; data curation, R.M., B.d.F. and A.D.P.; writing—original draft preparation, R.M. and A.D.P.; writing—review and editing R.M. and A.D.P.; taxonomic revision, A.D.P. All authors have read and agreed to the published version of the manuscript.

**Funding:** This research received no external funding.

**Institutional Review Board Statement:** Not applicable.

**Informed Consent Statement:** Not applicable.

**Data Availability Statement:** No new data were created in this study. Data sharing is not applicable to this article.

**Conflicts of Interest:** The authors declare no conflicts of interest.

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
