# Peer review of "Bryophytes Used in Folk Medicine: An Ethnobotanical Overview"

_horticulturae, doi:10.3390/horticulturae9020137_

Round 1

Reviewer 1 Report

The MS is rather interesting and useful. Please address further comments to improve it:

Line 41, 149, 151, 152, 153, 155, 156, 162, 164 Sphagnum should be italic

Lines 65-67. Sentence not clear. Reformulate

Line 108-110. and Table 1. Please use international two-letter country codes. This way it is confusing.

https://www.iban.com/country-codes

F. japonicas, Mnium cuspidatum within the Table 1. Should be italic

Philippine should be replaced to Philippines throughout the manuscript.

The authorities of species are present in Table 1, so if already mention in table 1, no needs to have it in the text as well. Please, remove.

Line 178. against should be non italic

Line 192. What does it mean dan?

Line 211. Species and should be non italic. Remove species authorities

Line 286. Not higher. Bryophytes are higher plants as well. You mean tracheophytes or vascular plants, isn’t it?

Author Response

We thank the Reviewer for the comments and suggestions that helped us to improve our manuscript.

Line 41, 149, 151, 152, 153, 155, 156, 162, 164 Sphagnum should be italic

DONE

Lines 65-67. Sentence not clear. Reformulate

According to the Reviewer comment the following sentence

Additionally, different bryophyte extracts and isolated compounds have demonstrate antimicrobial, antiviral, cytotoxic effects, and on smooth and non-striated muscles, and weight loss.

has been changed to:

Different compounds isolated from bryophytes have demonstrate antimicrobial, antiviral, neuroprotective and cytotoxic effects. Furthermore, they have shown their positive effects on smooth muscles, such as stomach, intestines, bladder, bronchioles, and uterus, playing also a role in the weight loss.

Line 108-110. and Table 1. Please use international two-letter country codes. This way it is confusing.

https://www.iban.com/country-codes

We thank the Reviewer for this important comment. Following the suggested web site, country codes have been changed throughout Table 1 and in the caption where, accordingly, the alphabetical order has also been updated.

  1. japonicas, Mnium cuspidatum within the Table 1. Should be italic

We thank the Reviewer for the suggestion but we prefer not to use italics for synonyms to avoid confusion

Philippine should be replaced to Philippines throughout the manuscript.

DONE

The authorities of species are present in Table 1, so if already mention in table 1, no needs to have it in the text as well. Please, remove.

DONE

Line 178. against should be non italic

DONE

Line 192. What does it mean dan?

We apologize for the typo, “dan” has been deleted

Line 211. Species and should be non italic. Remove species authorities

DONE

Line 286. Not higher. Bryophytes are higher plants as well. You mean tracheophytes or vascular plants, isn’t it?

According to the Reviewer suggestion "higher plants” has been changed to “vascular plants”

Reviewer 2 Report

The work is a very successful summary of knowledge about the use of mosses in a folk medicine. Based on the study of a considerable amount of literary sources, the authors created an overview of moss species with this specific uses. In addition, they show how these plants are used in different countries of the world, or in different cultures, traditions or medical schools.

I consider the manuscript to be very useful and it will certainly be sought after as a source of information for further studies of this issue.

I recommend accepting it for publication.

Author Response

We are very grateful to the Reviewer for his flattering comments

Reviewer 3 Report

The review is a complete overview of the uses of Bryophytes in the folk medicine, few minor corrections to be done, see file attached.

Well written.

Maybe could be added the  methodology of selection or elimination of documents in the mat and methods section, which is quite small

Author Response

We thank the Reviewer for the comments and suggestions that helped us to improve our manuscript.

Line 37

“organism” has been changed to “organisms”

Line 57

“they” has been changed to “their”

Line 102

Comma has been added. (6.1%) has been changed to (6.1% each)

Table 1

N.D. has been changed to N.A. and in the caption has been added (N.A.= Not Available)

Line 199

  1. Patents has been eliminated

Materials and methods

Following the Reviewer comment, the following sentences have been added in the M&M chapter

A simple evaluation of both title and abstract was carried out for every result in relation to the use of briophytes in folk medicine. The articles filtered in the previous point had their abstracts fully read in order to further reveal the real interest of the review article and to filter out non-applicable studies. The results thus obtained had their full contents read and evaluated. Only articles containing specific references to the use of briophytes were included.